# The Effect of Ketogenic Diet on Shared Risk Factors of Cardiovascular Disease and Cancer

**DOI:** 10.3390/nu14173499

**Published:** 2022-08-25

**Authors:** Noushin Mohammadifard, Fahimeh Haghighatdoost, Mehran Rahimlou, Ana Paula Santos Rodrigues, Mohammadamin Khajavi Gaskarei, Paria Okhovat, Cesar de Oliveira, Erika Aparecida Silveira, Nizal Sarrafzadegan

**Affiliations:** 1Isfahan Cardiovascular Research Center, Cardiovascular Research Institute, Isfahan University of Medical Sciences, Isfahan 8158388994, Iran; 2Interventional Cardiology Research Center, Cardiovascular Research Institute, Isfahan University of Medical Sciences, Isfahan 8158388994, Iran; 3Department of Nutrition, Faculty of Medicine, Zanjan University of Medical Sciences, Zanjan 4515863994, Iran; 4Health Care Superintendence, Goiás State Health Department, Goiânia 74093-250, Brazil; 5Heart Failure Research Center, Cardiovascular Research Institute, Isfahan University of Medical Sciences, Isfahan 8158388994, Iran; 6Pediatric Cardiovascular Research Center, Cardiovascular Research Institute, Isfahan University of Medical Sciences, Isfahan 8158388994, Iran; 7Department of Epidemiology & Public Health, Institute of Epidemiology & Health Care, University College, London WC1E 6BT, UK; 8Postgraduate Program in Health Sciences, Faculty of Medicine, Federal University of Goiás, Goiânia 74690-900, Brazil; 9Faculty of Medicine, School of Population and Public Health, University of British Columbia, Vancouver, BC V6T 1Z3, Canada

**Keywords:** ketogenic diet, cardiovascular disease, cancer, hypertension, oxidation, inflammation, diabetes, obesity

## Abstract

Cardiovascular disease (CVD) and cancer are the first and second leading causes of death worldwide, respectively. Epidemiological evidence has demonstrated that the incidence of cancer is elevated in patients with CVD and vice versa. However, these conditions are usually regarded as separate events despite the presence of shared risk factors between both conditions, such as metabolic abnormalities and lifestyle. Cohort studies suggested that controlling for CVD risk factors may have an impact on cancer incidence. Therefore, it could be concluded that interventions that improve CVD and cancer shared risk factors may potentially be effective in preventing and treating both diseases. The ketogenic diet (KD), a low-carbohydrate and high-fat diet, has been widely prescribed in weight loss programs for metabolic abnormalities. Furthermore, recent research has investigated the effects of KD on the treatment of numerous diseases, including CVD and cancer, due to its role in promoting ketolysis, ketogenesis, and modifying many other metabolic pathways with potential favorable health effects. However, there is still great debate regarding prescribing KD in patients either with CVD or cancer. Considering the number of studies on this topic, there is a clear need to summarize potential mechanisms through which KD can improve cardiovascular health and control cell proliferation. In this review, we explained the history of KD, its types, and physiological effects and discussed how it could play a role in CVD and cancer treatment and prevention.

## 1. Introduction

Cardiovascular disease (CVD) and cancer are the first and second leading causes of death worldwide, respectively [1]. CVD accounted for approximately 17.8 million deaths, followed by cancer with 9.56 million deaths in 2017 [1]. These conditions are usually treated as separate events. However, they share several risk factors, such as obesity, diabetes, hypertension, hyperlipidemia, diet, and lifestyle, suggesting some shared biology [2,3]. Among the pathophysiological mechanisms, many processes in common between CVD and cancer have been investigated, including inflammation, oxidative stress, resistance to cell death, cellular proliferation, neurohormonal stress, angiogenesis, and genomic instability [2,4]. Epidemiological evidence has demonstrated that the incidence of cancer was elevated in patients with CVDs, such as heart failure (HF). The opposite was also true, as an increased occurrence of HF in cancer survivors may be related to chemotherapy, radiotherapy, and immunotherapy, often combined [3]. 

Cohort studies suggested that controlling for CVD risk factors may also have an impact on cancer incidence and its outcomes. In the European Prospective Investigation into Cancer and Nutrition (EPIC) study, a large cohort of 23,153 individuals aged 35 to 65 years, adherence to a healthy lifestyle (no smoking, body mass index (BMI) < 30, physical activity > 3.5 h weekly, and healthy diet) resulted in a hazard ratio (HR) of 0.19 (95% confidence interval (CI), 0.07–0.53) for myocardial infarction; 0.50 (95% CI, 0.21–1.18) for stroke; and 0.64 (95% CI, 0.43–0.95) for cancer after 7.8 years of follow-up [5]. The Atherosclerosis Risk in Communities Study (ARIC), with 13,253 participants aged 45 to 64 years between 1987 to 2006, demonstrated that adhering to six of the seven ideal health metrics reduced the risk of incident cancer compared to subjects meeting zero ideal health metrics by 51% [6]. Therefore, interventions that improve CVD and cancer shared risk factors may play a role in the development of a combined prevention and treatment strategy for both diseases.

Several nutritional interventions have been tested to prevent and treat CVD and cancer [7,8,9,10]. The ketogenic diet (KD), which consists of a low-carbohydrate and high-fat diet, was developed in 1920 for the treatment of refractory epilepsy with successful outcomes and became widely known in the 1970s when used for weight loss purposes. KD has been recently investigated for the treatment of numerous diseases, including CVD and cancer, due to its role in promoting ketolysis, ketogenesis, and modifying many other metabolic pathways that might lead to beneficial health effects [11]. Considering the number of studies on this topic, there is a clear need to summarize the evidence on the potential therapeutic mechanisms of KD on CVD and cancer. This review presented a wide range of studies that examined the impact of KD on CVD and cancers and their shared risk factors and the potential mechanisms involved.

## 2. Overview of Ketogenic Diet Metabolism and Its Physiological Effects

### 2.1. History

Over the centuries, fasting has been used as the primary treatment for epilepsy [12]. In the early twentieth century, Guepa and Maria used fasting to treat epilepsy in France [13]. Ten years later, in 1921, Wilder suggested that this treatment for epilepsy may be due to body ketones [14] and hence introduced KD, which produced similar biochemical changes as fasting [15]. In the same year, Woodiat stated that in conditions of starvation or low carbohydrates, ketone bodies appeared in the blood [16]. KD consists of a significant amount of fats, low carbohydrates, and limited protein, which leads to altered energy metabolism and an increase in body ketones in the blood, which eventually forces the body to use them to produce energy [17,18]. The recommended ratio of fat to protein plus carbohydrates varies between 4:1 and 2:1 by weight [15]. Later, Wilder and Teperman described this diet as 1 g and 10–15 g per kilogram of body weight of protein and carbohydrates, respectively, and the rest of the required energy from fat [16]. Table 1 shows different versions of KD.

### 2.2. Classification

In general, four types of KD have been characterized. The first type is the classical or traditional type, with a 4:1 ratio of fat to protein plus carbohydrates. Thus, in this diet, fat intake provides 90% of the energy, which may be difficult and intolerant to the public [19]. The second type contains medium-chain-triglyceride (MCT) like caprylic acid, capric acid, caproic acid, and lauric acid [20], with 70% of energy uptake from fat including 10% long chain triglyceride (LCT) fat and 60% MCT fat, 20% from carbohydrate, and 10% from protein. Due to MCT’s faster membrane diffusion, the absorption process occurs earlier. Since the production rate of ketone bodies is higher than the previous type, for its equal formulation, less fat intake is required [21]. The third type of KD with a 1.1:1 ratio of 65% fat, 10% carbohydrate, and 25% protein was developed in 1970 by Dr. Robert Atkins based on carbohydrate restriction for weight loss. It is characterized by a higher carbohydrate restriction and high-fat intake with no restrictions on calories and protein. This diet was called the Atkins modified diet [22]. The fourth type is called the low glycemic index (GI) diet, which recommends food with a GI lower than 50 with fat to carbohydrate ratio of 6:1 [23].

### 2.3. Physiology and Metabolism

A drop in blood glucose levels through fasting or starvation stimulates the liver to produce glucose by breaking down glycogen stores and through gluconeogenesis [24]. Therefore, persistent low blood glucose leads to the preferential breakdown of fat, with a major contribution to increased lipolysis due to lowered insulin levels. However, under these conditions, energy metabolism can hardly continue beyond the production of acetyl-coenzyme A (CoA). This is due to the limited availability of oxaloacetate for oxidation in the tricarboxylic acid (TCA) cycle in hepatocytes. This way, the liver produces ketone bodies as metabolic energy for extrahepatic tissues. Ketosis is caused by the preferential breakdown of fats for energy production due to an insufficient amount of carbohydrates, causing low systemic insulin levels [25]. 

This process of ketone production is controlled by the sensitive lipase hormone, acetyl-CoA carboxylase, and 3-hydroxy-3-methylglutaryl (HMG) CoA synthase, all of which are regulated by three hormones: adrenaline, insulin, and glucagon [26]. By acting on lipase-sensitive hormones, glucagon leads to the production of fatty acids and thus increases ketogenesis while inhibiting acetyl-CoA carboxylase, which prevents fatty acids from entering the liver mitochondria and increases ketogenesis. In the process of ketogenesis, glucagon accelerates ketogenesis by stimulating HMG CoA synthase [27,28]. Unlike glucagon, insulin has an inhibitory effect on ketogenesis by inhibiting lipolysis and increasing lipogenesis, leading to a reduction in available fatty acids [27].

The amount of ketone bodies in KD is between 0.5 and 3 mmoL, which is called nutritional or physiological ketosis [29,30]. Reducing carbohydrate intake to less than 50 g per day and suppressing subsequent insulin secretion leads to reduced glucose and fat storage, increases the triacylglycerol lysis, and the formed fatty acids are subsequently converted to ketone bodies [29]. The glycogen of the liver and skeletal muscle is depleted during carbohydrate restriction for 24 h and one week, respectively, in the ketogenesis process to supply energy to the mitochondrial matrix of the liver [31,32]. The condensation of β-oxidation-derived acetyl-coenzyme (acetyl-CoA) into acetoacetate (AcAc) is initiated by the sequential need of mitochondrial enzymes 3-hydroxy-3-methylglutaryl CoA synthase and 3-hydroxy-3-methylglutaryl CoA lyase. Acetoacetate forms acetone through spontaneous decarboxylation, which is a volatile substance. By reducing the ketone fractions, it becomes beta-hydroxybutyrate, which is the circulating form of ketone bodies in the human body [33,34].

Ketolysis occurs in almost all extrahepatic tissues [35]. Once in the bloodstream, monocarboxylic acid carriers transport ketone bodies [36] and are absorbed by the brain and other tissues. The key enzyme in the ketolysis process is succinyl CoA transferase, which is the most active in the heart, muscle, kidneys, nervous system, and skeletal muscles [37]. First, β-hydroxybutyrate (BOHB) is converted back to AcAc by NAD+ oxidation. It is finally converted to acetyl-CoA by BDH1, succinyl-CoA:3-oxoacid CoA transferase (SCOT), and mitochondrial thiolase, which are used in the Krebs cycle [12]. Figure 1 and Figure 2 illustrate the process of ketogenesis and ketolysis in the ketogenic diet.

## 3. Ketogenic Diet and Cardiovascular Disease

### 3.1. Ketogenic Diet and Cardiovascular Disease Function

The effect of dietary intake on the cardiovascular system has been studied more than other physiological functions [38]. In recent studies, the therapeutic effect of KD in CVD has been investigated, especially in the case of heart failure with a reduced ejection fraction [39]. Many animal and human studies, both observational and experimental, indicated the effect of KD on cardiovascular health. A meta-analysis on 1141 obese patients demonstrated that KD had a beneficial effect on cardiovascular health [40]. Another randomized clinical trial (RCT) by Polito et al. [41] showed that the heart rate decreased following weight loss due to KD. Several mechanisms proposed the impact of KD on cardiovascular function. 

### 3.2. Ketogenic Diet and Energy Inducing in Heart

Ketone bodies have cardio-protection effects in patients with heart failure with a reduced ejection fraction from improvements in the cardiac metabolic state and may specifically increase cardiac efficiency [42]. However, a few animal studies have indicated that increasing BOHB by hyperketonemia had no effect on improving cardiac efficiency in diabetic mice [43].

Mobilization of fatty acids, glucose, lactate, ketones, and amino acids provides a cardiac energy source. Recent studies showed the role of ketone bodies as an important fuel source for the heart [42]. Ketone bodies are a source of energy during fasting, starvation, uncontrolled diabetes, heart failure, prolonged exercise as well as following KD consumption. Another potential mechanism for the beneficial effect of KD on CVD is the relationship between carbohydrate intake and CVD. A recent meta-analysis of 19 cohort studies including 15,663,111 participants suggested that higher carbohydrate intake was directly associated with increased CVD and stroke risk, while no association was found for coronary heart disease (CHD) or CVD mortality [44]. However, some studies showed conflicting findings. Lagiou et al. found that low carbohydrate–high protein diets, used on a regular basis and without consideration of the nature of the carbohydrates or source of proteins, were associated with an increased risk of cardiovascular events in women [45].

### 3.3. Ketogenic Diet and Endothelial Function

A ketone body of 1,3-butanediol (BD) has some advantages in experimental models of hypoxia or ischemia. BD is transformed to BOHB, which is an influential endothelium-dependent vasodilator through increasing nitric oxide (NO) synthase and hence may have a beneficial role in CVD as KD-fed animals had increased endothelial NO synthase protein expressions [46]. Myocardial blood flow was enhanced by BOHB infusion by about 75% and induced vasodilatation [42]. Cellular senescence is a process that eventually leads to an irreversible state of growth arrest. In addition, there is also much evidence that shows old cells have adverse effects on the body like tissue regeneration, aging process, vascular diseases, and endothelial dysfunction cells [47]. Since aging is a CVD risk factor, delaying or preventing it can play a main role in cardiovascular dysfunction. Caloric restriction with the production of ketones can prevent the aging of vascular cells [48]. Improvement in progressive dilation of the heart and cardiac contractile function was observed in a study following KD in mice [39]. Studies have demonstrated that myocardial function may be improved under the influence of ketone bodies following KD [49]. It has been suggested that a low level of ketone bodies through KD exerts beneficial effects on inflammatory status, senescence, and metabolism of endothelial cells. However, hyperketonemia in patients with diabetes-induced the inflammatory status, which led to the adhesion of monocytes to endothelial cells and endothelial dysfunction that consequently increased CVD risk [49]. 

### 3.4. Ketogenic Diet and Mitochondrial Function

One potential mechanism related to the beneficial role of KD on CVD is its effect on mitochondrial function. It has been suggested that mitochondrial dynamics play an essential role in heart function. Although diabetes decreases the mitochondrial respiratory control ratio and adenosine triphosphate (ATP) content, which can break the mitochondrial membrane and reduce their size, KD improved mitochondrial dynamics by preventing mitochondrial fission and hence raised its function in mice [50]. The mitochondria played an important role in controlling energy levels in cells and ATP production through oxidative phosphorylation and electron transport chain (ETC) activity in an in vitro study [51]. Thus, KD may protect against heart damage by regulating the heart’s energy metabolism and mitochondrial function. 

### 3.5. Ketogenic Diet and Inflammation

In most vascular diseases, atherogenesis is a common underlying cause, and inflammation plays a role in the atherogenesis process. It is possible that the low level of ketone bodies can help in reducing inflammation in the vessels of individuals with CVD [49]. BOHB has an anti-inflammatory effect on the heart by suppressing the Nod-like receptor protein 3’s (NLRP3’s) inflammasome expression. This multiprotein complex has an important role in cardiac inflammation, as seen in myocardium dysfunction in rats [52]. In addition, BOHB inhibits pro-inflammatory cytokine production [52]. Thus, the anti-inflammatory action of ketone bodies is a potential beneficial cardio-protective mechanism of KD.

### 3.6. Ketogenic Diet and Oxidative Stress

Reactive oxygen species (ROS) or reactive nitrogen species (RNS) play a crucial role in the cell signaling pathways and regulate cellular function [53]. However, an imbalanced redox state, mitochondrial dysfunction, and inefficient antioxidant system can lead to overproduction of ROS or RNS. Elevated levels of these factors contribute to the pathogenesis of various diseases, which is mediated through their possible damaging effects on the cell membrane, protein, and DNA [53]. ROS is a by-product of normal metabolism in the mitochondrial respiratory chain [54]. Therefore, in cells with high mitochondria density, like cardiac myocytes, higher oxygen consumption predisposes them to oxidative stress. Increments of ketone body oxidation through BOHB de-hydrogenases 1 overexpression increase the antioxidant superoxide dismutase production and reduces the oxidative stress following transaortic constriction in rats [55]. However, owing to few human studies and a rather low number of studies in animal models, the beneficial effects of nutritional ketosis on the recovery of ischemia cannot be concluded [39,56].

### 3.7. Ketogenic Diet and Carotid Intima-Media Thickness

Carotid intima-media thickness (cIMT) has been used as an indicator of cardiovascular disease risk. Several studies have demonstrated that there was no significant change in cardiac indexes, including cIMT and elastic properties of the carotid artery, aortic strain, the stiffness index, and distensibility after short- and long-term KD in epileptic children [57,58]. Kapetanakis et al. reported that one year of KD treatment decreased the carotid artery distensibility. However, there was no improvement in cIMT and carotid artery compliance. In addition, the beneficial changes of KD on the cardiovascular system did not remain after two years [40].

In conclusion, although KD has controversial effects on the cardiovascular system, many studies have revealed a potential role of KD in CVD prevention, treatment, and disease reversal through improvements in energy induction, endothelial function, mitochondrial function, inflammation state, and antioxidant effect.

## 4. Proposed Mechanism of Action of Ketogenic Diet in Cancer

### 4.1. Glucose Dependence on Cancer Cells

Various mechanisms have been proposed for the inhibitory effects of KD on the growth of tumors and cancer cells. Part of these inhibitory effects is through KD’s effect on glucose metabolism in cancer cells. Elevated glucose uptake is a frequent feature of cancer cells, which is used to produce mediators of lipid, protein, and nucleic acids synthesis [59]. Hyperglycemia and hyperinsulinemia have been identified as stimuli for tumor growth in several cancers, such as breast, pancreatic, and colorectal [60]. In fact, cancer cells use glycolysis as their main pathway of energy production to have more access to glucose and speed up glucose utilization, even in the presence of adequate amounts of oxygen, resulting in the accumulation of lactate, a process called the Warburg effect [61]. For this reason, it seems that slowing down the Warburg effect and reducing access to glucose by administering low-carbohydrate diets and KDs to cancer patients may have positive effects on slowing the growth of cancer cells [62]. It has been reported in several pre-clinical studies that KD administration led to a significant reduction in serum glucose levels. For example, Woolf et al., in an animal study of 21 mice models of glioma, demonstrated that the administration of a high-fat, low-carbohydrate, adequate protein ketogenic diet led to a significant reduction in blood glucose, both 7- and 14-days post-implantation [63]. Lussier et al., in another animal study of ten mice models of malignant glioma, demonstrated that mice fed a therapeutic KD for 7 days had a significant reduction in blood glucose [64]. Another study that was conducted on mice models of medulloblastoma showed that serum insulin and glucose levels were significantly reduced in mice fed KD for 7 days [65]. Moreover, Caso et al., in an animal study of 160 male SCID mice (mice with severe combined immunodeficiency), showed that mice fed a no-carbohydrate ketogenic diet (NCKD) for 14 days had significantly lower glucose levels [66]. KD not only reduced the serum concentration of glucose but also reduced the serum levels of insulin and insulin-like growth factor 1 (IGF1), which are involved in tumorigenesis, and this can have a synergistic effect [62]. Tsujimoto et al., in a population-based study on 9778 participants aged over 20 years, showed that insulinemia was an important risk factor for cancer death in individuals with and without obesity [67]. Carbohydrate intake restriction, followed by decreased insulin levels in patients on low-carbohydrate diets such as the KD, can partially suppress these pathways in cancer cells [68].

Fine et al., in a pilot clinical trial of 10 patients with advanced cancer, showed a significant inverse correlation between ketone bodies serum levels and insulin and IGF1 concentration [69]. Similar results were reported in another human study of 73 women with ovarian cancer [70]. Activation of insulin receptors following hyperglycemia-induced the activity of some enzymatic pathways and caused cascades such as the RAS-mitogen activated protein (RAS-MAP) pathway that plays an important role in the growth and proliferation of cancer cells. In fact, activating the RAS-MAP pathway stimulated mitogenic effects of insulin and also induced cell survival by protein kinase B (Akt) and mechanistic target of rapamycin (mTOR), and nuclear factor kappa-light-chain enhancer of activated B cells (NF-κB) which activated antiapoptotic and pro-inflammatory pathways [71]. In addition, high serum insulin levels have been shown to stimulate the gene expression of vascular endothelial growth factor (VEGF), which is an essential component of vascular angiogenesis [72]. 

### 4.2. Mitochondrial Metabolism and Cancer

Another mechanism involved in the beneficial effects of KD on cancer cells through its effect on mitochondrial metabolism. 

Numerous experimental studies have shown that in some cancers, such as cancers of the head and neck [73], liver, breast, colon, ovary, and prostate, mutations occur in mitochondrial DNA and impair normal mitochondrial function. One of the main reasons for this mutation is the increase in the level of ROS and oxidative stress in cancer patients, which causes mitochondrial damage and mutations [74,75]. Since, in cancer cells, the oxidative phosphorylation (OXPHOS) of mitochondria is not performed well, mitochondrial mass decreases, and its function is impaired, cancer cells use aerobic fermentation to supply their own energy. Replacing ketone bodies with carbohydrates as an energy source in cancer cells requires the use of efficient mitochondria to provide the energy needed for these cells to grow and proliferate. Considering the low efficiency of mitochondria in these cells, the presence of ketone bodies cannot stimulate the growth of cancer cells like carbohydrates [76]. 

### 4.3. Oxidative Stress and Cancer Cells

It has been reported in a review study that ROS can activate metabolic adaptations, the phosphoinositide 3-kinase (PI3K) and the hypoxia-inducible factor (HIF) pathways and consequently change bioenergetic metabolisms and cause tumorigenicity [77]. 

Another proposed mechanism of KD on cancer cells is increasing the level of oxidative stress in these cells. Firstly, KD reduces the amount of glucose available to produce glucose 6-phosphate, pyruvate, and NADPH production, which is necessary for reducing hydroperoxides. On the other hand, cancer cells do not have efficient mitochondria, and shifting the energy supply pathway to mitochondrial metabolism and disrupting the ETCs function resulted in increased one-electron reductions of O_2_ leading to ROS production, and is exposed to cancer cells under conditions of oxidative stress. In fact, when glucose is restricted in the diet of cancer patients, cancer cells are expected to be exposed to more oxidative stress, which in turn leads to more apoptosis of cancer cells and slower growth and proliferation [18,78]. The difference between high-fat and high-protein diets is that although high-protein diets also shift energy to mitochondrial metabolism, some amino acids contribute to NADPH production by entering the citric acid cycle and cannot increase oxidative stress levels in the tumor environment as much as high-fat diets such as KD. The effectiveness of KD on the level of oxidative stress in cancer cells has been investigated by various experimental studies [62,79]. Allen et al., in an animal study, found that KD administration 2 days before radio-chemotherapy in a mouse model of lung cancer improved the effects of radio-chemotherapy responses by enhancing oxidative stress [80].

### 4.4. Ketogenic Diet and Systemic Inflammation

One of the main factors involved in the etiology and progression of various cancers is systemic inflammation [81]. Systemic inflammation reduces the feeling of hunger, exacerbates cytokine cascades, activates catabolic pathways, induces lean body mass loss, induces cancer-induced wasting syndrome (cachexia) in cancer patients, and reduces the quality of life [82,83]. On the other hand, obesity, one of the main risk factors for cancer, has also been shown to increase the risk of the disease by exacerbating systemic inflammation [84]. Under normal circumstances, regulatory mechanisms in the body control the amount of inflammation. One of the important regulatory mechanisms in immune cells is mediated by proteins called inflammasomes, which are classified into two groups based on structural features: nucleotide-binding and oligomerization domain (NOD)-like receptors (NLRs) and absent melanoma 2 (AIM2)-like receptors (ALRs) [85]. One of these important proteins (a mutation in this protein is associated with inflammatory diseases) is NLRP3. Some of the previous review studies have shown that therapeutic approaches that inhibit the expression of this protein can reduce the growth of cancer cells and prolong survival [86,87]. 

It has been reported that ketone metabolites, specifically β-hydroxybutyrate, can directly suppress NLRP3 and its related cytokine cascades [88]. Other experimental studies have shown that the administration of KD in animal models inhibited NLRP3 [50,89,90]. In fact, KD stimulates the production of β-hydroxybutyrate and AcAc in the liver, which inhibits some of the factors involved in the development of cancer cells, such as NLRP3. Moreover, it has been found that KD has anti-inflammatory, anti-angiogenic, and anti-invasive functions that could kill cancer cells by pro-apoptotic and anti-inflammatory mechanisms [68].

### 4.5. Ketogenic Diet in Combination with Chemotherapy and Radiotherapy

Schmidt et al. designed the first study, which evaluated a low carbohydrate ketogenic diet (LCKD) during chemotherapy in 2011. In this study, sixteen patients with advanced metastatic tumors and no conventional therapeutic options with adherence to LCKD (less than 70 g/day of carbohydrates) were evaluated in terms of their quality of life. It was found that LCKD during chemotherapy for three months was suitable, and there were no severe side effects and even some improvements in quality of life [91]. It has additionally been reported that KD was safe and effective in patients with high-grade glioma during treatment [92]. An animal study of neuroblastoma cases revealed that KD plus chemotherapy exerted anti-tumor effects, suppressing growth and causing a significant reduction of tumor blood-vessel density and intra-tumoral hemorrhage, accompanied by activation of AMP-activated protein kinase in neuroblastoma cells [93]. Allen et al. evaluated the effects of KD plus radiation as well as conventionally fractionated radiation combined with carboplatin in lung cancer xenografts. They found that KD plus radiotherapy reduced tumor growth by a mechanism that may involve increased oxidative stress [80]. Moreover, Yang et al., in an animal study, reported that KD administered for 30 days in mice models with pancreatic cancer increased the sensitivity of cancer cells to chemotherapy and disrupted pancreatic cancer cell metabolism and growth [94]. A prospective clinical trial from Japan evaluated the effects of KD with fluorouracil-based chemotherapy compared to a control group among patients with rectal cancer. They found that patients who had adhered to KD had a higher response rate to chemotherapy than the control group (60% in the intervention vs. 21% in the control group) [95]. Moreover, the results of a clinical study in patients with pancreatic ductal carcinoma showed that patients who have been treated with gemcitabine or FOLFIRINOX chemotherapy revealed that those patients in the KD group experienced a median overall survival of 15.8 months (95% CI 10.5–21.1) compared to the group that received only chemotherapy and hyperthermia only [96]. 

Despite the positive results observed, the results of some human studies are contradictory. Woodhouse et al., in a clinical study with 29 patients, evaluated the effects of KD plus temozolomide and radiation on tumor progression six months after the completion of radiotherapy on magnetic resonance image (MRI) scans. They did not find any significant correlation between β-hydroxybutyrate and tumor progression [97]. In addition, in another clinical study in eleven patients with glioblastoma, it was reported that the overall survival in patients ranged between 9.8 and 19.0 months, which was not a significant improvement. Quality of life, neurological functioning, and Karnofsky Performance Scale outcomes did not change substantially during the study [98]. 

### 4.6. Indication and Contraindication of Ketogenic Diet

Over the past few decades, the effectiveness of KD on cancer cells has been evaluated in both pre-clinical and clinical studies. One of the types of cancers that has been associated with positive results in pre-clinical studies of KD is brain cancer. Morscher et al., in a pre-clinical study on 11 mice models with neuroblastoma tumors, showed that KD administered for three weeks with or without calorie restriction led to a significant reduction in tumor growth and prolonged survival [99]. Aminzadeh-Gohari et al., in another study, found significant positive results after the administration of KD with medium-chain triglycerides for 40 days on neuroblastoma xenografts in a CD1-nu mouse model [93]. Glioblastoma is another type of cancer that has been studied extensively in relation to KD. Champ et al., in a retrospective review of patients with high-grade glioma, found that KD was well tolerated in most of the patients and led to a significant reduction in serum glucose levels and increased survival [100]. In a case report study of women aged 65 years, it was reported that two months of adherence to KD led to a 20% reduction in body weight, but no brain tumor tissue was detected [100]. Like the pre-clinical studies, most of the human studies on KD were conducted on patients with brain tumors. In contrast to the safe application of KD reported in various cancer models, there were some concerns about KD use in some cancer types. Vidali et al., in an animal study on 10 CD-1 nu/nu mice models of renal cell carcinoma with features of Stauffer’s syndrome, showed that KD administered for two weeks led to severe weight loss and liver dysfunction [101]. Liśkiewicz et al., in an animal study that evaluated the effects of long-term KD treatment on kidney cancer, showed a pro-tumor effect of this diet that aggravated the disease [102]. Another worrying result was seen in models of mice with BRAF V600E-positive melanoma whose KD exacerbated tumor growth. In fact, they found that acetoacetate as a ketone metabolite could induce BRAF V600E mutant-dependent mitogen-activated protein kinase 1 (MEK1) activation in melanoma cells, and KD might play a pathogenic effect in this cancer [103].

Furthermore, some short-term side effects (such as hypercholesterolemia, dehydration, constipation, acidosis, lethargy, and gastrointestinal distress) [104,105] and long-term side effects (such as kidney problems, cardiomyopathy, hyperlipidemia, and bone mineral loss) following the administration of KD in cancer patients were reported [68]. It is worth mentioning that KD may cause deficiencies in some micronutrients in patients, and patients who are exposed to nutritional deficiencies may exacerbate their nutrition status if they adhere to such diets [106].

## 5. Effects of Ketogenic Diet on Shared Cardiovascular and Cancer Aspects: Pre-Clinical and Clinical Studies

### 5.1. Ketogenic Diet and Oxidative Stress

Despite several studies exploring the effects of KD on oxidative stress, findings are still controversial. In a rat model study, KD reduced oxidative damage (protein nitration, 4-Hydroxynonenal (4-HNE) adducts, and 8-hydroxydeoxyguanosine (8-OHdG)), but it increased antioxidant defenses (glutathione peroxidase (GPx) and superoxide dismutase (SOD)) activity [107]. It is assumed that KD increases mitochondrial biogenesis, which, in turn, induces cellular resistance to oxidative stress at both genetic and mitochondrial levels [108]. In contrast, KD might exert anti-tumor effects through increasing intratumor oxidative stress and inducing apoptosis against tumor cells [80,109]. These effects were mainly attributed to lipid peroxidation [110]. In addition, while KD diminished plasma SOD levels in Wistar rats after two months [111], it might have improved mitochondrial respiratory complex activity and increased the production of some antioxidants such as glutathione and detoxification enzymes [55,112]. There is evidence that BOHB, as a ketone body, mitigates ROS production and improves mitochondrial respiration [113]. However, there is no consensus in this context, and further studies are required to demonstrate how ketone bodies affect redox signaling molecules and pathways.

### 5.2. Ketogenic Diet and Inflammation

The effect of KD on inflammatory biomarkers is inconsistent and may differ from one tissue to another [114]. For example, 4 weeks’ consumption of KD in mice induced inflammation and macrophage accumulation in the liver while decreasing them in white adipose tissue [114]. In a human model study, a 4-week isocaloric KD, containing 80% energy from fat, increased C-reactive proteins but not interleukin-6 in comparison with a control diet [115]. These findings were in line with other studies [116,117] but not all [118]. Nevertheless, in diabetic patients, a low carbohydrate diet (20% energy from carbohydrates) may be preferred over a low-fat diet in terms of subclinical inflammatory improvement, though both diets were similarly effective in weight reduction [118]. A recent systematic review of 63 studies assessing the impact of KD on inflammatory biomarkers showed that in approximately three-quarters of these studies [119], inflammatory biomarkers were reduced after KD. KD might exert an anti-inflammatory property via various mechanisms, including inhibiting activation of the nuclear factor kappa-light-chain-enhancer of activated B cells and the inflammatory nucleotide-binding, leucine-rich-containing family, pyrin domain-containing-3, and histone deacetylases [113]. 

### 5.3. Mitochondrial Function of Ketogenic Diet

Mitochondrial bioenergetics dysregulation and elevated ROS production are associated with different metabolic diseases. There is mounting evidence suggesting that KD can affect these processes. KD stimulates mitochondrial biogenesis and increases uncoupling proteins in the hippocampus of animals [112,120]. ROS production is regulated by uncoupling proteins. Furthermore, the reduced effect of β-hydroxybutyrate and AcAc on ROS production may be another reason for modified mitochondrial functions in KD. Rats fed KD for 3 weeks had higher mitochondrial reduced glutathione to oxidized glutathione ratio (GSH:GSSG) compared with the control group, indicating an improvement in GSH production. Lower production of H_2_O_2_ in KD-fed rats elicited an improvement in the mitochondrial functions [112]. KD increased mitochondrial density by regulating the deacetylation of various mitochondrial proteins [121]. The KD diet may benefit mitochondrial function via enhancing antioxidant activity, reducing ROS production, and ROS-induced damages [122].

### 5.4. Microbiota and Epigenetics: Therapeutic Approaches in Ketogenic Diet

Gut microbiota, as a “hidden metabolic organ” or a “virtual endocrine organ” [123], interacts with diet and has an enormous influence on the host metabolic pathways and health status. To date, little is known about the effect of KD on gut microbiota, and most of the information comes from trials on patients with neurological or mental disorders. It has been shown that the source and the proportion of dietary fat [124] and protein [125] have diverse effects on gut microbiota. For example, in patients with obesity, KD containing whey or vegetable protein, in comparison with an animal-based KD, led to a greater decline in the *Firmicutes* to *Bacteroidetes* ratio [125]. This ratio goes up in individuals with and is associated with various comorbidities [126]. Notably, KD reduced microbiota diversity, but its associated weight loss can increase microbiota diversity in the long term [127,128]. Other changes in gut microbiota following KD include a growth in the abundance of *Alistipes* and *Parabacteroides*, which are negatively correlated with waist circumference, and a decline in the abundance of *Orodibacter splanchnicus* and *Lactobacillus* [129]. In a 6-month pre- and post-intervention on refractory epilepsy patients, KD decreased the richness of gut microbiota, *Firmicutes*, and *Actinobacteria* but increased *Bacteroidetes* [130]. Further research to clarify the long-term effects of KD, with different macronutrient compositions and sources, is warranted. 

### 5.5. Effect of Ketogenic Diet on Cardiovascular and Cancer Shared Risk Factors

Overall, owing to poor adherence to any restrictive diet [131], the long-term effects of KD on various metabolic risk factors are not well-established. In addition, KD composition varies from one study to another. For instance, restricted-carbohydrate diets might be either high in fat or protein and differ in their fatty acids and protein sources. In addition, the type of saturated fatty acid (SFA) might be a determinant of health outcomes [132,133], and the ratio of different dietary fatty acids can potentially affect health consequences [134,135]. Meta-analyses reported no significant association between SFA and CVD [136], type 2 diabetes [137], or lipid profile [138,139]. Therefore, these factors should be considered when interpreting the effects of KD. Table 2 shows the summary of studies that assessed the shared mechanisms between cancer and cardiovascular diseases.

#### 5.5.1. Hypertension

Previous meta-analyses have shown that hypertension is a risk factor for different cancers, such as breast, prostate, colorectal, and renal [144,145,146]. Mutual pathophysiological pathways mediated by adipose tissue, such as elevated inflammatory biomarkers, can increase the risk of hypertension, cancers, and CVD. However, hypertension may block apoptosis and disrupt cell turnover [145]. Hypertension modifies the activity of the sympathetic nervous system (SNS) and thereby affects the risk of both cancer and CVD through androgen-mediated pathways [146]. Besides the beneficial effects of KD on reducing adipose tissue [147,148], KD can regulate SNS activity by reducing salivary cortisol and the hypothalamus-pituitary-adrenal (HPA) axis [149]. In addition, β-hydroxybutyrate suppresses SNS activity by antagonizing G protein-coupled receptor 41 (GPR41) [150] and reducing heart rate as an indicator of SNS activity [41]. 

#### 5.5.2. Dyslipidemia

The contributory role of dyslipidemia in CVD and some specific cancers is well documented [151,152,153]. Dyslipidemia increases cancer and CVD risk through some shared mechanisms, such as their effects on insulin resistance, inflammation, and oxidative stress [154,155,156,157]. Therefore, improving lipid profile can be a viable approach to reducing both CVD and cancer risks. A recent meta-analysis suggested that KD decreased serum levels of total cholesterol, low-density lipoprotein cholesterol (LDL-C), and triglyceride and increased high-density lipoprotein cholesterol (HDL-C) in patients with type 2 diabetes [158]. However, another meta-analysis on overweight and obese individuals showed a reduction only in triglyceride and total cholesterol following KD consumption [147]. The follow-up duration of these studies ranged from 1 to 56 weeks in diabetic patients [158] and 3 to 104 weeks in overweight and obese individuals [147]. Another meta-analysis of long-term (>1 year) effects of KD, in comparison with a low-fat diet, showed a greater reduction in triglyceride but increased LDL-C and HDL-C [148]. Differences in the composition of KD can influence its effects. Furthermore, it has been shown that KD increases the particle size of LDL-C, which has lower atherogenicity compared with smaller ones [159]. Therefore, improvement of the lipid profile because of KD might be associated with a lower risk of cardiomyocyte dysfunction and impede carcinogenicity processes. However, the fatty acid composition of KD is a relevant factor that can modify KD’s effect on the lipid profile. Traditional KD is high in saturated fats, which come from cream (up to 50% of fat intake), butter, bacon, and other proteins high in SFA [160]. Hyperlipidemia is a common consequence of traditional KD, while substituting SFA with unsaturated fatty acids can prevent hyperlipidemia even with fat consumption of up to 90% of daily energy intake [160]. A KD with an appropriate composition of protein, mono-unsaturated and saturated fatty acids can increase HDL-C and reduce triglycerides, LDL-C, body fat mass, and inflammatory biomarkers [106,140].

#### 5.5.3. Obesity

Obesity, especially abdominal obesity, is a well-known risk factor for both CVD and cancer [161]. According to the results of a meta-analysis, longer adherence to the KD leads to more weight loss, which can be kept off even two years after follow-up [147]. In comparison with a low-fat diet, KD is also associated with a greater than 0.9 kg weight reduction in the long term [148]. Castellana et al., in a systematic review and meta-analysis study, showed that very-low-calorie KD (VLCKD) with less than 800 kcal energy intake and 50 g carbohydrate administered to overweight and obese individuals led to a significant reduction in BMI and waist circumference [147]. In a RCT study among patients with ovarian or endometrial cancer, it was reported that after 12 weeks, patients in the KD group, compared to the control group, had lower fat mass, body weight, and fasting serum insulin. Also, they found a significant inverse association between the changes in serum β-hydroxybutyrate and insulin-like growth factor 1 (IGF-1) concentrations [70]. Increased levels of IGF-1 were associated with several common cancer types because IGF-1 has mitogenic and antiapoptotic effects [162]. Similar findings were reported for waist circumference. In overweight and obese individuals, KD could decrease waist circumference by 12.6 cm [147]. Excess body fat, in particular abdominal adiposity, is linked to higher inflammation and oxidative stress, which play a crucial role in the pathogenesis of both cancer and CVD [53]. Gut microbiota might be another link between obesity and cancer. On the other hand, obesity is associated with an increased abundance of mollicutes, a subclass of *Firmicutes*. The higher fecal mollicutes content, the higher the production of lactate, acetate, and butyrate. This, in turn, promotes fat storage and causes obesity. Gut microbiota also contributes to nutrient metabolism and may metabolize them into mutagens, such as N-nitroso compounds and heterocyclic amines [163]. Indeed, altered gut microbiota in obese patients may affect tumor initiation and development as well as change inflammatory processes [163]. There is evidence suggesting that KD can decrease *Firmicutes* and increase *Bacteroidetes* [164]. However, it is interesting to note that gut flora activity is also influenced by environmental factors such as diet. In a murine model study, transmitted bacteria derived from lean twins to mice were less protective against obesity when they were fed with a high saturated fatty acids diet compared with a low-SFA diet [163]. These findings highlight the relevance of a KD composition to obtain benefits.

#### 5.5.4. Diabetes Mellitus

Hyperinsulinemia and hyperglycemia are associated with an increased risk of CVD and cancer [2]. Insulin affects lipid metabolism pathways such as lipid uptake, lipolysis, and lipogenesis and is potentially associated with obesity [165]. It also affects the neoplastic process through its stimulating effects on mitogenesis, proliferation, invasion, and metastasis [11]. Increased levels of free IGF-1 because of hyperinsulinemia have mitogenic and antiapoptotic properties, which can enhance cancer promotion and progression [11]. In addition, a U-shaped association has been reported between IGF-1 and nonfatal CVD events in elderly men [166]. A meta-analysis of RCTs indicated beneficial effects of a KD on glycemic control, with a greater improvement in diabetic patients [167]. Improvement in glycemic parameters was obtained after 2 weeks [168] and continued throughout the long term (56 weeks) [141], which is probably mediated by the inverse association between circulating levels of ketone bodies and hepatic glucose output [11]. Given that hyperinsulinemia mediates the adverse effects of diabetes on both cancer and CVD, improving insulin resistance can be an advantageous strategy to prevent their progression. In addition, weight and fat mass loss are closely correlated with insulin sensitivity. An RCT on newly diagnosed patients with type 2 diabetes showed that a periodic KD could reduce fasting insulin more than a healthy diet [142]. In another trial, depending on the adherence rate, 70–90% of patients with diabetes who initially presented on insulin and followed a low-calorie diet with carbohydrate restrictions were no longer taking insulin after one year [143]. 

## 6. Conclusions

Several potential mechanisms have been proposed to explain the potential effects of KD on cancer and CVD. KD might suppress oxidation through increasing mitochondrial biogenesis, exert an anti-inflammatory effect and consequently reduce cancer and CVD risk, and improve various shared risk factors of cancer and CVD (e.g., hypertension, obesity, diabetes mellitus, and dyslipidemia). However, it has been well-established that the effects of KD might vary by its composition and type of macronutrients. Although available evidence has shown that KD can improve shared CVD and cancer risk factors over a short period of time, there are only a few long-term trials and this limits the strength of recommendations for KD. Moreover, there is no study that has exclusively examined KD effects on patients with both medical conditions, and therefore the interplay between KD and various pathophysiological pathways involved in both conditions cannot be exactly known. In addition, adherence to such a restrictive diet is difficult for children and older adults, and even adults for the long term. Therefore, an alternative diet might be a more appropriate approach to managing metabolic abnormalities. It was suggested that future studies examine KD effects in people from different age groups, follow participants and their adherence rate for a longer period, and compare the efficacy of KD with other healthy diets (e.g., Dietary Approaches to Stop Hypertension (DASH) or Mediterranean diet) to improve various risk factors.

## Figures and Tables

**Figure 1 nutrients-14-03499-f001:**
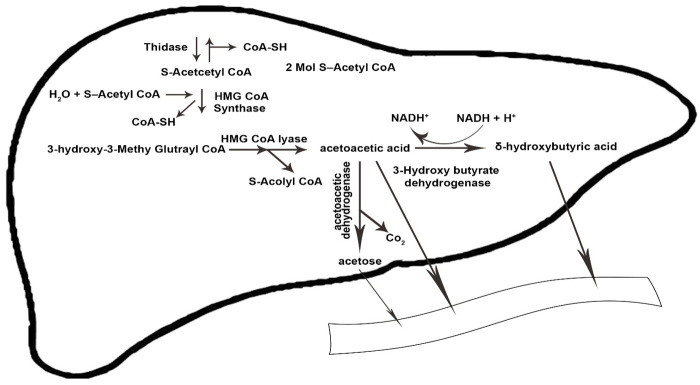
The ketogenesis process in the body. CoA-SH, coenzyme A-SH; CoA, coenzyme A; H_2_O, water; HMG-CoA, 3-hydroxy-3-methylglutaryl coenzyme A; NADH^+^, nicotinamide adenine dinucleotide + hydrogen; H^+^, Hydrogen; Co_2_, carbon dioxide. The arrows indicate the direction of changes in the process, as well substrates and products.

**Figure 2 nutrients-14-03499-f002:**
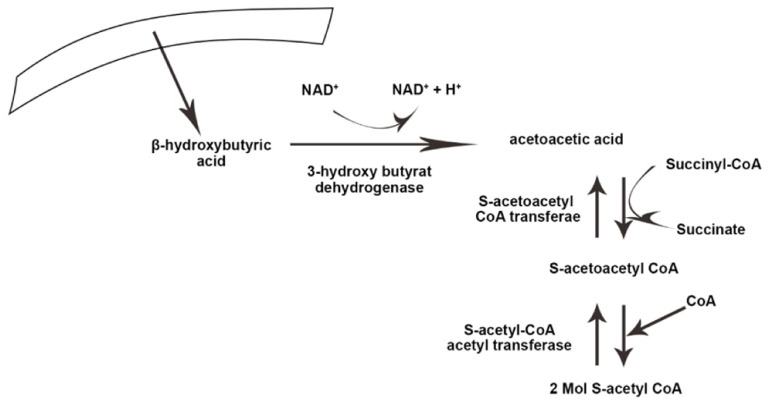
The process of ketolysis in the body. NAD^+^, Nicotinamide adenine dinucleotide. The arrows indicate the direct ion of changes in the process, as well substrates and products.

**Table 1 nutrients-14-03499-t001:** Different types of ketogenic diet (KD).

KD Type	Macronutrient Proportion (% of Total Energy)			General Characteristics
	Carbohydrate	Fat	Protein	
Classic ketogenic diet	4	90	6	Developed for epilepsy treatment
Medium-chain-triglyceride (MCT) ketogenic diet	17	73 (30–60% MCT)	10	MCT supplements should be incorporated into all meals and snacks
The modified Atkins diet (MAD)	5 (10–20 g/day)	65	30	No restriction on energy content, fluid, or protein
The modified ketogenic diet (MKD)	5 (30 g/day)	65–80	20–25	No restriction on energy
Very low-calorie ketogenic diet (VLCKD)	13 (usually <30 g/day)	44	43 (1.2–1.5 g/kg of ideal body weight)	Total energy intake of <800 kcal/day
Ketogenic Mediterranean diet/modified Mediterranean ketogenic diet	<30–50 g/day	45–50	30–35	With an emphasis on lean meats, fish, olive oil, walnuts, and salad

**Table 2 nutrients-14-03499-t002:** Summary of studies that assessed shared mechanisms between cancer and cardiovascular diseases [70,115,125,127,130,140,141,142,143].

First Author, Year, Country	Population	Study Design	Follow Up	Intervention	Comparator	Outcomes
Wolver S, 2021, United States	85 T2DM who initially presented on insulin the age 56.1 ± 9.9 years 70% Female	One arm intervention	1 year	LCKD (20 g of total carbohydrates per day from non-starchy vegetables)	-	↓ insulin dose, HbA1c, and weight
Li S, 2022, China	60 overweight or obese patients newly diagnosed with T2DM	RCT	12 weeks	KD (the main foods for the diet were olive oil, butter, fried eggs, double-fried pork, pan-fried salmon, pacific saury, sardines, broccoli, avocado, etc.; daily limits for ingredients were as follows: carbohydrate 30–50 g, protein 60 g, fat 130 g, and total calories 1500 kcal + 2000 mL water/day	Routine diet for diabetes (carbohydrate 250–280 g, protein 60 g, fat 20 g), total calories (1500 kcal, without no limitation on foods) + 2000 mL water/day	Greater ↓ in weight, BMI, waist, TG, TC, LDL-C, HDL-C, FBG, FIN, and HbA1c in the KD group compared with the control group
Dashti HM, 2006, Kuwait	66 healthy obese patients (BMI ≥ 30 kg/m^2^) with a high cholesterol level (group I; *n* = 35) and normal cholesterol level (group II; *n* = 31)	Non-randomized clinical trial	56 weeks	KD (less than 20 g of carbohydrates in the form of green vegetables and salad, and 80–100 g of proteins in the form of meat, fish, fowl, eggs, shellfish, and cheese. PUFA and MUFA (5 tablespoons olive oil) were included in the diet.	KD (less than 20 g of carbohydrates in the form of green vegetables and salad and 80–100 g of proteins in the form of meat, fish, fowl, eggs, shellfish, and cheese. PUFA and MUFA (5 tablespoons olive oil) were included in the diet.	↓ weight, BMI, TC, LDL-C, TG, FBG, and ↑ HDL-C in both groups. KD was safe and beneficial in both groups.
Volek JS, 2009, USA	40 subjects with atherogenic dyslipidemia	RCT	12 weeks	Low carbohydrate diet % carbohydrate: fat: protein = 12:59:28	Low-fat diet (56:24:20)	Greater ↓ in glucose and insulin levels, insulin sensitivity, weight, adiposity, and more favorable TG, HDL-C, and TC/HDL-C ratio in the low carbohydrate diet group compared with the low-fat diet group
Zhang Y, 2018, China	20 patients (14 males, 6 females)	Single arm trial	6 months	KD non-fasting diet with a classic 4:1 ratio KD fat:protein plus carbohydrates. Fat from pork, oils such as olive oil, coconut oil, and other sources. Plant fat accounted for about 70% of the amount of fat. At least 1 g of protein per kg body weight from animal sources (e.g., eggs, meat, poultry, and fish). Carbohydrate-containing foods such as fruits and vegetables were added.	-	↓ alpha diversity in fecal microbiota ↓ abundance of *Firmicutes* ↑ levels of *Bacteroidetes*
Gutiérrez-Repiso C, 2019, Spain	33 obese patients	RCT	2 months	VLCKD + synbiotics	Low-calorie diet	↔ microbial diversity ↑ short-chain fatty acid-producing bacteria ↑ *Odoribacter* and *Lachnospira*
Basciani S, 2020, Italy	48 patients with obesity (19 males and 29 females, HOMA index ≥ 2.5, aged 56.2 ± 6.1 years, BMI 35.9 ± 4.1 kg/m^2^	RCT	45 days	VLCKD regimens (≤800 kcal/day) containing whey, plant, or animal protein		Greater ↓ in relative abundance of *Firmicutes* and greater ↑ in *Bacteroidetes* in whey and plant protein compared with animal protein group.
Rosenbaum M, 2019, USA	17 men (BMI: 25–35 kg/m^2^)	Single arm trial	4 weeks	Isocaloric KD (15% protein, 5% carbohydrate, 80% fat)	Baseline diet (15% protein, 50% carbohydrate, 35% fat)	↑ free fatty acids, TC, LDL-C, and CRP ↓ Fasting insulin, C-peptides, TG, and fibroblast growth factor 21
Cohen CW, 2018, Birmingham	women with ovarian or endometrial cancer (age: ≥19 years; BMI ≥ 18.5 kg/m^2^)	RCT	12 weeks	KD (70:25:5 energy from fat, protein, and carbohydrate)	The American Cancer Society diet (high-fiber, low-fat)	Greater ↓ in total and android fat mass, visceral fat, and fasting serum insulin in KD compared with control ↔ lean mass

T2DM: diabetes mellitus type II; LCKD: low-carbohydrate ketogenic diet; HbA1c: hemoglobin A1c; RCT: randomized clinical trial; KD: ketogenic diet; BMI: body mass index; TG: triglyceride; TC: total cholesterol; LDL-C: low-density lipoprotein cholesterol; HDL-C: high-density lipoprotein cholesterol; FBG: fasting blood glucose; FIN: fasting insulin; PUFA: polyunsaturated fatty acid; MUFA: mono-unsaturated fatty acid; VLCKD: very-low-calorie ketogenic diet; HOMA: homeostatic model assessment; CRP: C-reactive protein; ↓: decrease; ↑: increase; ↔: no change.

## Data Availability

Not applicable.

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
