# Peer review of "The Effect of Ketogenic Diet on Shared Risk Factors of Cardiovascular Disease and Cancer"

_nutrients, 2022, doi:10.3390/nu14173499_

Round 1

Reviewer 1 Report

In this manuscript the Authors review the literature that examined the effects of KD on CVD and cancers.

The manuscript is well written and comprehensively summarizes current knowledges on these topics, even if the informations provided are not very original.

I only suggest a minimal revision in the Introduction:

Line 89; please, specify that ratio of fat to protein + carbohydrates is expressed in grams

Line 90:  4:1 instead of 4;1.In this manuscript the Authors review the literature that examined the effects of KD on CVD and cancers.

Author Response

Reviewer 1

Comments and Suggestions for Authors

In this manuscript the Authors review the literature that examined the effects of KD on CVD and cancers. The manuscript is well written and comprehensively summarizes current knowledges on these topics, even if the informations provided are not very original.

 I only suggest a minimal revision in the Introduction:

Line 89; please, specify that ratio of fat to protein + carbohydrates is expressed in grams.

Thanks. We declared that this ratio is by weight (page 2, line 84).

Line 90:  4:1 instead of 4;1.In this manuscript the Authors review the literature that examined the effects of KD on CVD and cancers.

Agreed. Corrected (page 2, line 84).

Reviewer 2 Report

Very interesting work and topic.

Reminders:

1.      technical-

·         missing numbering-e.g. line 155, 413,415..

·         double citation- 39=53, 49=50

2.      scientific content-

·         give a clear table of different types of KD diets ·         distinguish pure low carb and KD throughout the article ·         state the type of studies - animals, humans ·         comment the number of study patients and length of studies ·         cite original citations (for instance 139) ·         line 160- RCT with 1 141 patients does not correspond to the given citation ·         citation 42-  the patients in the study are not exclusively diabetic ·         citation 45- specify the conclusion of the study  ( Low carbohydrate-high protein diets, used on a regular basis and without consideration of the nature of carbohydrates or the source of proteins, are associated with increased risk of cardiovascular disease)   ·         line 214-217- a bold claim with no support in the literature and in the topic ·         line 238- „following transaortic constriction is not true- A CCI injury was produced on the exposed left cortex using an electronically controlled pneumatic piston cylinder (Hydraulics Control, Inc., Emeryville, CA, USA) as previously described.26 In the present study, the 5 mm diameter flat rod tip was angled at 22.5° away from vertical and compressed the cortex at 1.9 m/s to a depth of 2 mm. After injury, a small piece of gelfoam was placed over the craniotomy site to reduce bleeding and the wound sutured closed. Sham-injured rats received only a craniotomy but no cortical injury. ·         Line 268- citation 64-67- ?? reduction of glukose levels?? For instance citation 64-Tumors from animals maintained on KD showed reduced expression of the hypoxia marker carbonic anhydrase 9, hypoxia inducible factor 1-alpha, and decreased activation of nuclear factor kappa B. Additionally, tumors from animals maintained on KD had reduced tumor microvasculature and decreased expression of vascular endothelial growth factor receptor 2, matrix metalloproteinase-2 and vimentin. Citation 65. We demonstrate that mice fed the KD had increased tumor-reactive innate and adaptive immune responses, including increased cytokine production and cytolysis via tumor-reactive CD8+ T cells. Additionally, we saw that mice maintained on the KD had increased CD4 infiltration, while T regulatory cell numbers stayed consistent. Lastly, mice fed the KD had a significant reduction in immune inhibitory receptor expression as well as decreased inhibitory ligand expression on glioma cells. ·         Paragraphs line 415-460 are not studies connecting exclusively CVD and cancer aspects

Author Response

Reviewer 2

Comments and Suggestions for Authors

Very interesting work and topic.

 Thanks very much.

Reminders:

  1. technical-
  • missing numbering-e.g. line 155, 413,415.

Thanks. Numbering was corrected when they had been missed (Pages 3-5, 9-11, lines 149, 150, 160, 180, 201, 213, 224, 239, 434, 436, 453, 504, 531).

  • double citation- 39=53, 49=50

Thank you. References were corrected for duplications.

  1. scientific content-
  • give a clear table of different types of KD diets· 

Thank you. Table 1 was added to describe different versions of KD (page 2, lines 86-87).

distinguish pure low carb and KD throughout the article ·        

Thank you. It has been corrected through the manuscript.

state the type of studies - animals, humans ·  

Thank you. It has been corrected through the manuscript.

comment the number of study patients and length of studies ·    

Thank you. It has been corrected through the manuscript.

cite original citations (for instance 39) ·       

Thank you. It was replaced (page 4, line 154).   

line 160- RCT with 1 141 patients does not correspond to the given citation. 

It was revised to: “A meta-analysis on 1,141 obese patients demonstrated that KD had a beneficial effect on cardiovascular health” and relevant reference was added (page 4, line 155-156).       

citation 42-  the patients in the study are not exclusively diabetic.      

Thank you. It was revised to “Ketone bodies have cardio-protection effects in patients with heart failure and reduced ejection fraction by improvements in the cardiac metabolic state and may specifically increase the cardiac efficiency” (page 4, lines 161-163)  

citation 45- specify the conclusion of the study (Low carbohydrate-high protein diets, used on a regular basis and without consideration of the nature of carbohydrates or the source of proteins, are associated with increased risk of cardiovascular disease).    

Thank you. This comment was implemented. “However, some studies showed conflicting findings. Lagiou et al. found that low carbohydrate-high protein diets, used on a regular basis and without consideration of the nature of carbohydrates or the source of proteins, are associated with increased risk of cardiovascular events in women.”  (page 4, lines 175-177).  

line 214-217- a bold claim with no support in the literature and in the topic ·   

Thank you. It was deleted.       

line 238- „following transaortic constriction is not true- A CCI injury was produced on the exposed left cortex using an electronically controlled pneumatic piston cylinder (Hydraulics Control, Inc., Emeryville, CA, USA) as previously described.26 In the present study, the 5 mm diameter flat rod tip was angled at 22.5° away from vertical and compressed the cortex at 1.9 m/s to a depth of 2 mm. After injury, a small piece of gelfoam was placed over the craniotomy site to reduce bleeding and the wound sutured closed. Sham-injured rats received only a craniotomy but no cortical injury.

Thank you. This part was revised to “However, owing to few human studies and rather low number of research in animal models, the beneficial effects of nutritional ketosis on recovery of ischemia cannot be concluded.” (page 5, lines 235-237).

Line 268- citation 64-67- ?? reduction of glukose levels?? For instance citation 64-Tumors from animals maintained on KD showed reduced expression of the hypoxia marker carbonic anhydrase 9, hypoxia inducible factor 1-alpha, and decreased activation of nuclear factor kappa B. Additionally, tumors from animals maintained on KD had reduced tumor microvasculature and decreased expression of vascular endothelial growth factor receptor 2, matrix metalloproteinase-2 and vimentin. Citation 65. We demonstrate that mice fed the KD had increased tumor-reactive innate and adaptive immune responses, including increased cytokine production and cytolysis via tumor-reactive CD8+ T cells. Additionally, we saw that mice maintained on the KD had increased CD4 infiltration, while T regulatory cell numbers stayed consistent. Lastly, mice fed the KD had a significant reduction in immune inhibitory receptor expression as well as decreased inhibitory ligand expression on glioma cells.

Thanks for your comment. The results of the mentioned references on glucose levels were summarized in this part given that this section is about “Glucose dependence of cancer cells” (page 6, lines 268-277).         

Paragraphs line 415-460 are not studies connecting exclusively CVD and cancer aspects.

Agreed. Due to the lack of studies which exclusively connect CVD and cancers, we summarized studies which separately assessed these two conditions but with the same pathophysiological mechanisms. This matter addressed clearly in page 12, lines 620-623.

Round 2

Reviewer 2 Report

all comments have been accepted